# Olfactory Evaluation in Clinical Medical Practice

**DOI:** 10.3390/jcm11226628

**Published:** 2022-11-08

**Authors:** Květoslav Červený, Karla Janoušková, Kristýna Vaněčková, Šárka Zavázalová, David Funda, Jaromír Astl, Richard Holy

**Affiliations:** 1Department of Otorhinolaryngology and Head and Neck Surgery, University Hospital Motol, First Faculty of Medicine, Charles University, 15006 Prague, Czech Republic; 2Institute of Anatomy, First Faculty of Medicine, Charles University, 12800 Prague, Czech Republic; 3Department of Otorhinolaryngology and Maxillofacial Surgery, Military University Hospital, 16902 Prague, Czech Republic; 4Third Faculty of Medicine, Charles University, 10000 Prague, Czech Republic; 5Laboratory of Cellular and Molecular Immunology, Institute of Microbiology of the CAS, v. v. i., 14220 Prague, Czech Republic

**Keywords:** odourants, olfactory system, anosmia, hyposmia, parosmia, objective olfactometry, olfactory event-related potentials

## Abstract

The subjective and demanding nature of olfactory testing means that it is often neglected in clinic despite loss of smell leading to significant limitations in everyday life. The list of diseases associated with loss of olfaction far exceeds the field of otorhinolaryngology and can also be seen in neurodegenerative disorders. Knowledge of possible clinical testing is essential to determine a proper differential diagnosis for the loss of olfactory sense. Causes of olfactory impairment can be divided into either failure in transferring odour to the organ of perception or damage to the olfactory pathway structure itself. Examination should therefore include methods evaluating cross-sectional area and patency of the nasal cavity as well as subjective or objective assessment of olfactory function. In this report we summarize several articles, studies, and our own experiences to provide a comprehensive review of their current clinical usage including their benefits, limitations, and possible outcomes. We also discuss the mechanism of olfaction step by step to provide a full understanding of the possible errors depending on the localization in the pathway and the methods designed for their detection. We discuss the correlation of the microbiome in nasal polyposis and chronic rhinitis with olfactory impairment using objective olfactometry. The topic of objective olfactometry and the examination of olfactory event-related potentials (OERP) is commented upon in detail.

## 1. Introduction

The aim of this article is to provide a closer look at the current methods of olfactory assessment and provide an algorithm that could be used in clinical practise as a part of a routine otorhinolaryngological examination. Smell is one of the basic senses by which we explore the world, although unlike in other mammalian animals, its function is limited in humans. Proper function of this sense is essential to human behaviour personally, socially, and professionally. Olfactory loss in connection with upper respiratory tract diseases is a relatively common finding in the otorhinolaryngology clinic, but it can also occur in connection with neurological, genetic, or metabolic diseases. Therefore, correct diagnosis can be essential for further diagnostic considerations. Perception of the olfactory stimuli relies on conduction; i.e., the delivery of the odourant stimulus to the olfactory region and the sensorineural processing in the olfactory organ itself.

## 2. Nasal Mucosa

There are two types of epithelium in the nasal cavity; one is called the respiratory part, and the other is considered as the olfactory part. The respiratory part is made of a thicker grey–pink pseudostratified columnar epithelium with cilia. It contains scattered goblet and seromucinous cells that cover the surface of the nasal mucosa by a mucus layer formed by three superimposed layers: the periciliary, the mucosal superficial layer, and the surfactant layer located between previous two. Its complete replacement takes 20–30 min thanks to the cilia of the ciliated epithelial cells vibrating towards the nasopharynx [1]. In addition to its immunological and biophysical functions, the mucus layer also plays an important role in the enzymatic conversion of inhaled substances, thus influencing the final pattern of olfactory perception [2]. The olfactory part is located on the ceiling of the nasal cavity at the site of the cribriform plate of the ethmoid bone, the adjacent part of the septum, superior nasal concha, and the middle nasal concha. It occupies an area of almost 5 cm^2^ [3]. The olfactory epithelium is made up of three types of cells: supporting, basal, and olfactory receptor cells (bipolar neurons) [4]. Olfactory cells have a remarkable capacity for regeneration as long as the basal cell layer is untouched and there is no damage significant enough to leave scar tissue that prevents the reconnection of axons (see Figure 1) [5,6].

## 3. Physiology and Neurophysiology of the Olfaction

Perception of the olfactory stimuli relies on conduction, i.e., the delivery of the odourant stimulus to the olfactory region and the sensorineural processing in the olfactory organ itself. The olfactory organ is divided into a peripheral part (olfactory epithelium and fila olfactoria) and a central part (bulb, olfactory tract, and centres in the brain).

### 3.1. Peripheral Part of the Olfactory Organ

Processing of sensory information begins in the nasal cavity on the dendrites of olfactory cells. Around 6–20 cilia protruding from their apical end are in contact with the mucus layer. Their surface is covered by odourant receptors (ORs). We distinguish approximately 350 different types of these receptors. These differ in the sequence of several amino acids, which allows considerable variability of the odourant binding site. Activation of several specific groups of ORs leads to the creation of a unique formula for a given substance. This formula is relayed by olfactory cell axons for processing in the upper structures. Interestingly, even at this level there is a relatively effective attenuation of the signal of the long-acting odour stimulus (olfactory fatigue or adaptation). The exact mechanism of this desensitization is not yet fully understood; however, reduction in quantity of ORs on the surface of cilia has been discussed as a possible mechanism [7,8,9].

### 3.2. Central Part of the Olfactory Organ

Axons of olfactory neurons pass into the olfactory bulb, ending in rich arborization at synapses with mitral and tufted cells. These structures, referred to as olfactory glomeruli, can be found in one bulb in a number of approximately 40 to 50. Each axon of an olfactory neuron carrying information from one type of ORs ends up in only one of such glomeruli. Mitral and tufted cell dendrites are arranged in the same manner. This significantly reduces the number of neurons carrying the signal from the olfactory analyser. Moreover, each glomerulus is simultaneously in contact with the periglomerular GABAergic cell. Thus, excitation signals coming from olfactory cells are modulated (by inhibition on dendrodendritic synapses with mitral and tufted cells) [7,8,9]. Granular cells are also found in the deeper layers of the bulb. They receive excitation from the secondary dendrites of mitral and tufted cells and reciprocally inhibit them. This makes it possible to decrease these glomeruli with a weak response to the stimulus and thus prevent information from being distorted before entering the cortex. Information which is processed and modified as mentioned above is transmitted by the axons of mitral and tufted cells through the olfactory tract. It is divided into medial, lateral, and intermediate olfactory stria and ends without interconnection directly in the primary olfactory areas (piriform cortex, olfactory nucleus et tubercule, amygdala, entorhinal cortex) and after interconnection in secondary olfactory areas (hippocampus, hypothalamus, thalamus, orbitofrontal cortex) (see Figure 2). In these structures, smell is involved in processes related to memory, emotions, mood, and other components of the central nervous system (CNS) [7,8]. On the other hand, efferent fibres from the cortex can adjust the output of the olfactory bulb according to the overall mood of the organism (e.g., when hungry) [9].

### 3.3. Trigeminal Afferentation

Trigeminal nerve fibres play a role in the perception of irritating stimuli (such as ammonia, ethanol, menthol, CO_2_, capsaicin, etc.). There is not much information about the mechanism of function of these trigeminal receptors. However, their irritation releases neuropeptides that cause swelling, pain, sneezing, or salivation leading to nasal obstruction and secretion [10]. Intact olfactory function has been shown to be necessary for proper trigeminal afferent function. These two pathways communicate with each other at the central level, and in patients with impaired olfactory nerve function, the response from the trigeminal nerve is reduced [11].

## 4. Classification of Olfactory Disorders

According to the latest International Classification of Diseases (11th Revision)/ICD-11/of the World Health Organization (WHO), there are the following code diagnoses: MB41.0 AnosmiaMB41.1 ParosmiaMB41.Z Disturbances of smell and taste, unspecified

In general, smell disorders are divided according to their symptomatology and etiopathogenesis. Quantitative disorders can be described as a reduced ability to detect odours (hyposmia) or the complete loss of smell (anosmia). Increased perception of odours (hyperosmia) is not a disorder in the true sense of the word. It is more of a symptom (as in migraines). Qualitative disorders are described as odour distortion (parosmia), olfactory hallucinations (phantosmia), and the inability to detect certain substances (specific anosmia) (Table 1).

Olfactory dysfunction is common. Population estimates suggest that 19.1% of adults suffer from loss of smell, a figure that rises to 80% in patients over the age of 75 [12].

Many studies have demonstrated that testing of odour identification becomes possible at about age 3, suggesting that children’s linguistic functioning is sufficiently mature at this age. Age of the subjects is also important factor, as age influences the number of identified odours. Several works of testing olfaction in children have been made. Common consent is that age between 3 and 4 years brings relevant results. Willingness to cooperate, tendency to distraction, number of known odour substances, or maturation of the olfactory organ are major possible factors limiting the examination in younger ages [13]. There is consensus that, compared to adults, smell loss is relatively uncommon in children. A recent analysis of over 1200 consecutive patients presenting with chemosensory complaints revealed that children 16 and under represented less than 2% of the patients [14].

Congenital olfactory disorder is described in the literature. This is a congenital disorder in which the patient has no sense of smell from an early age, with an incidence of approximately 1 in 10,000, with the most common being Kallmann syndrome. Isolated congenital anosmia (ICA) patients show neurophysiologic deficits and some anatomic differences compared with healthy controls. The absence of olfactory event-related potentials (OERP) combining with a depth of olfactory sulcus less than 8 mm (Magnetic resonance imaging /MRI/ scans of olfactory pathway) is the important indicator for clinical diagnosis of ICA. The structure of the olfactory bulb may be a critical factor for clinical classification of ICA [14,15].

In clinical practice, it is favourable to sort disorders by their aetiopathogenesis, which allows us to localize the pathology more precisely and is also helpful in targeting subsequent treatment. We can distinguish between conductive and sensorineural disorders. In conductive disorders, the odourant cannot reach the area of the olfactory region (e.g., mucus layer structural changes, nasal polyps) or there is impaired ventilation of the nasal cavity (obturation of choanae by polyps/adenoid vegetation or after laryngectomy). Sensorineural disorders develop when the peripheral part of the olfactory organ (viral anosmia, chemical damage of the olfactory epithelium) or central part (traumatic conditions) is damaged (Table 2.)

## 5. Diseases Related to Olfactory Disorders

Altered or impaired olfactory function is a symptom involving several disease states across medical disciplines. Although most of these diseases belong to the otorhinolaryngology portfolio, the differential diagnosis should not be neglected, and data presented in the patient history should be considered as well. Emphasis should be placed on the circumstances leading to the disorder, time and duration, accompanying symptoms and medical history, as well as information regarding social background and job. The most common causes of olfactory disorders are sinonasal diseases. Inflammation-related increase in perfusion of mucosal capillaries in acute rhinosinusitis results in blocking of the olfactory region. In chronic rhinosinusitis, the blockage is usually caused by nasal polyps (see Figure 3) (even under physiological conditions, only about 10% of inhaled air passes through the olfactory region during calm breathing). In the study conducted by our department of otorhinolaryngology, we observed a significant improvement in smell in patients after primary pansinus surgery [16]. 

We can report from our own experience of olfactory disturbance in a patient with olfactory meningioma (see Figure 4).

Viral diseases cause prolonged olfactory loss, most likely due to a protective response of neurons to prevent the spread of the virus intracranially [17,18].

Current discussion regarding anosmia caused by COVID-19 seems to suggest the loss of smell is caused by viral damage to the olfactory nerve during its entry through the Angiotensin converting enzyme-2 (ACE2) receptor and transmembrane serine protease 2. Persistent post-viral olfactory disorders are estimated at 30% of patients 1 year after COVID-19 infection. No treatment is, to date, significantly effective on persistent post-viral olfactory disorders with the exception of olfactory training [18]. Qualitative olfactory dysfunctions after COVID-19 infection have been recognized as affecting mood, food enjoyment, reducing ability to detect dangers, influencing health status, and impacting social life. Moreover, in reports of post-viral smell alteration, studies have found that as many as 56% of patients experience parosmia and phantosmia. These are recognized as having a particularly pronounced impact on quality of life as most experiences involve unpleasant smells (malodours) [19].

Another common cause of smell impairment is cranio-trauma. This is due to shearing of fila olfactoria passing through the lamina cribrosa (e.g., during contusion), anterior cranial fossa fracture or external nose and nasal cavity fracture (due to mucosal oedema, submucosal bleeding, nasal bone fractures, collapse of the nasal septum after repeated trauma and subsequent scarring in the nasal cavity) [20]. Damage to the olfactory epithelium by toxic substances can be seen, for example, in chemical industry workers. Interestingly, when orthonasal perception is impaired by toxic substances, retronasal perception remains partially unharmed. One theory to explain this is that there is different vulnerability of the anterior and posterior parts of the olfactory epithelium to damage [21]. Anosmia can also be seen as an early symptom of Alzheimer’s disease and can appear as early as the mild cognitive deficit stage. This allows to distinguish the second common cause of dementia, vascular dementia, in which the olfactory disorder occurs later [22]. Parkinson’s disease can also manifest itself in the early stages by loss, reduction, or impairment of olfactory discrimination, regardless of the classic symptoms of the disease [23]. Other causes include olfactory disorders due to diabetes mellitus [24], age-related loss of smell (atrophy of nerve structures and epithelium, reduction of mucosal blood flow, reduction of foramina cribrosa, etc.) [25], or smoking [26]. Smell (and taste) functions are also impaired in patients with Sjögren’s syndrome due to a reduced amount of mucus that transports odourants and is due to recurrent upper respiratory tract infections [27]. Noteworthy is the presence of anosmia in granulomatosis with polyangiitis (formerly Wegener’s disease), due to chronic purulent secretion together with septal defects and collapse of nasal structures [28] or impaired olfactory function in patients with inflammatory bowel diseases [29].

### Nasal Microbiome and Olfactory Disorder

Chronic rhinosinusitis (CRS) and nasal polyps (NP) are common and recurrent diseases in otorhinolaryngology (ENT) practice. It is very often associated with olfactory impairment. The pathogenesis of CRS with or without NP and their association with allergies remain unclear, as does the exact mechanism of olfactory disturbance. Optimal treatment that also prevents recurrences is lacking. Epithelial cell dysregulation, pro-inflammatory chemokines and cytokines, mechanisms of natural immunity, and very recently described changes in the nasal microbiome represent important areas for further research on CRS and NP [30]. There is our project (NU 22-09-00493), aiming to identify new phenotypes or subphenotypes of chronic rhinosinusitis with or without NP by combining detailed mapping of the composition of the nasal microbiome with immunological parameters (cytokine/chemokine profiles, changes in immune cell populations) to optimize therapies or secondary prevention. Another aim of this project is to correlate the microbiome in nasal polyposis and chronic rhinitis with olfactory impairment using objective olfactometry.

## 6. Clinical Methods of Olfactory Assessment: Our Algorithms

The algorithm for diagnosing olfactory disorders includes examination of the nasal cavity and the olfactory organ itself. Before using subjective and objective methods (Table 3) it is appropriate to evaluate the airflow in the nasal cavity. Physical examination (disfiguration of the external nose and palpation of its bony and cartilaginous structures) and instrumental examination (such as anterior rhinoscopy and rigid or flexible endoscopy) are commonly used for assessment. Acoustic rhinometry (AR) and rhinomanometry (RMN) (see Figure 5) can be used to objectively evaluate the patency and cross-sectional area and length of the nasal cavity.

AR uses a principle that is not very different from the more commonly understood ultrasound. The acoustic signal emitted by the examination tube is reflected from the walls of the nasal cavity and returned to the microphone in the tube for analysis and processed into a graphic record. With a known initial intensity of the signal transmitted by the tube, the change in the magnitude of the reflected signal is directly proportional to the cross section of the specific part of the nasal cavity. In addition, the exact distance can be measured based on the time delay. The output is a graph on which the vertical axis corresponds to the distance from the nasal entrance (in centimetres), and the horizontal axis corresponds the minimal cross section (in cm^3^). The accuracy of this method is comparable to magnetic resonance imaging measurements [31].

RMN simultaneously measures nasal flow and pressure in nasopharynx, either through a probe inserted directly into the nasopharynx (posterior RMN) or through a contralateral nasal passage (anterior RMN). This method can further be divided into active or passive according to the patient’s participation in the examination. Active assessment is performed with the patient breathing through the nose with a mask. In a passive assessment there is no patient participation in breathing. In clinical practice the active anterior RMN is most commonly used. The patient breathes with a face mask and the pressure is measured through the bilateral nostril. The resistance is mainly affected by the different volumes of the cavity because the airflow in the mask is known and the length of the nasal cavity remains constant. Valid results can be achieved mainly by comparing the results before and after the decongestion test [31].

### 6.1. Subjective Methods

The results of subjective methods can be influenced by the patient himself. The main principle of these methods is to present the odourous substance and evaluate its interpretation by the patient. Usually, the odourous substance is presented directly by sniffing the carrier filled with odourant, i.e., orthonasally and birinally.

Subjective methods are divided based on odourant concentration into threshold and above-threshold. Threshold methods detect the lowest possible concentration of a substance (most often phenethyl alcohol or n-butanol) that the patient is able to detect. Supra-threshold methods present the substance in such concentrations that allow the subject to either distinguish individual odourants—discrimination—or to correctly identify and name them—identification. These methods are thus focused on quantitative rather than qualitative disorders. They cannot reliably prove the severity of isolated olfactory nerve damage with preserved trigeminal nerve function. Frequently used tests include the Sniffin’ Sticks test, the Odorized Markers Test (OMT), and the University of Pennsylvania Smell Identification Test (UPSIT) [32,33].

The Sniffin Sticks test kit contains 16 odourant-impregnated pen-like odour dispensing devices which should be identified by the patient (see Figure 6) [32,33].

The advantages of this test are its speed, simplicity, and the ability to test the smell threshold (odour detection threshold). Disadvantages include the short expiration and therefore greater costs of this method. A more affordable, and likewise time-saving, variant is OMT in the form of impregnated coloured markers. This method was developed and put into practice by Czech medical specialists [32]. In the first part of this test, after sniffing, the patient first spontaneously names offered odourous substances and in the second part selects the most suitable variant from the four offered choices. Points are given for each correctly named substance. The last test is UPSIT, which in the form of a small brochure presents to the patient 40 scratch and sniff strips. Scent is released from the microgranules using a pencil. This test is the most time consuming and is only single use [33,34,35].

### 6.2. Objective Methods

The outcome of objective methods is independent of the will of the patient. These are represented mainly by objective olfactometry. It is based on the principle of presenting the odourant by a special device (olfactometer) into the patient’s nasal cavity and detecting the odourant-evoked electric activity of neurons (synaptic activity) in the olfactory pathway. The major advantage is the objectification of the response to the odourous substance and straight assessment of the preserved function of the olfactory nerve. This also helps in detecting patient’s malingering. However, obtaining valid results is not easy and many aspects, not only technical aspects, need to be taken into account [33,36].

The olfactometer itself (see Figure 7 and Figure 8) works on the principle of dilution of the odourant in clean, odourless, unpolluted air, presented to the edge of the patient’s nasal vestibule by a tube. All internal components of the device must be made of materials that prevent contamination by other odours. It is also advisable to place the olfactometer in a quiet and well-ventilated room. To measure the potentials of both nerves (olfactory and trigeminal nerve) it is necessary to use substances that selectively stimulate only one of them. Therefore, vanillin, which selectively stimulates the olfactory nerve and CO_2_ for the trigeminal nerve, are used as odourants. The odourant is dissolved in a liquid (most often distilled water), through which air is bubbling, creating an aerosol. This provides adequate humidity, preventing drying the nasal mucosa during the experiment as well as maintaining constant temperature and preventing unpleasant and undesirable trigeminal response. When choosing a solvent, its physicochemical properties must be considered. Different pH values or direct interactions between the liquid and the odourant can significantly change the perception, and thus the results cannot be valid. The resulting aerosol with odourant must then be properly humidified (≥80%), heated to a temperature close to the body temperature (36 °C/96.8 F), and administered in a constant flow rate (8 L/min). Dry, too cold/warm air, or a higher flow rate irritates the trigeminal terminal fibres, and the result is the summation of signals from the trigeminal and olfactory fibres [33]. This also occurs during physiological changes in airflow in the nasal cavity (again by stimulation of trigeminal fibres). Therefore, the subject must breathe through the mouth throughout the experiment and the mixture is delivered only by the device (see Figure 8). Subsequently, the corresponding study protocol is selected in the computer program. Duration of each individual stimuli, the intervals between them and the scheme in which order the odourant (or more odourants) and CO_2_ are presented can be set. The aim is to choose a sequence in which there is minimal risk of habituation to the odourant (changing of odours and CO_2_), minimal duration of the stimulus to provide sufficient results, and, of course, minimal harm to the patient [33]. The response to the stimuli itself can be measured in several ways: by measuring the negative potential of the mucosa, classical electroencephalography (EEG), or by MRI.

Simultaneous activation of the ORs group leads to depolarization. The negative potential can be measured by electrodes directly placed on the nasal mucosa at the site of highest concentration of these receptors or at the site of trigeminal terminals. This method of measurement finds its application in animal models. In humans, suitable sites for electrode placement are not yet fully discovered and determining its proper position on the mucosa would need to be correlated with the picture of potentials typical of ORs [33,36]. The considerable invasiveness of this procedure is a significant disadvantage. For this reason, EEG is more commonly used. The waves of cortical stimuli processing are recorded and then evaluated (see Figure 9). During this scan, artefacts are caused by muscle contractions (the blinking caused by an unexpected stimulus for instance). An electrode that measures the muscular activity of the orbicularis oculi muscle must be applied to filter them out.

To distract the patient and obtain valid results, a simple computer application is also used (e.g., to keep the circle inside the moving square with the personal computer (PC) mouse), and the patient can listen to white noise [36,37] using the headphones [36,37].

The result of the examination is a curve of olfactory event-related potentials (OERP). The basis of the method in practice is most often the use of EEG and the output is an EEG recording during stimulation of the olfactory nerve with an odourant. Simultaneous activation of a group of olfactory receptors (OR) leads to depolarization, the negative potential of which can be monitored most easily in practice by EEG, where a cortical stimulus processing curve is plotted and subsequently evaluated. The three most important peaks N1, P2, P3 of their latency and amplitude are evaluated and the N1–P2 interval is assessed. The absence of olfactory potentials is a strong indicator of the presence of olfactory dysfunction [36,37].

## 7. Discussion

### 7.1. Discussion on Subjective Methods

In general, these methods are sufficient for a tentative olfactory assessment and their results can be used as a guide for further differential diagnostic steps. They are commonly used in clinical practice due to their simplicity, easy reproducibility of results, speed, and low purchase price. Their application highly depends on the choice of a set of odourous substances with which the population is familiar and for which normative values are created. Unknown odours, such as root beer or turpentine in the US version of UPSIT [34,35], can lead to misinterpretation. For correct results, however, it is necessary to determine the normative values of the healthy Czech population.

### 7.2. Discussion on Objective Olfactometry

Objective olfactometry is a method whose results demonstrate the functionality of both the olfactory and the trigeminal nerve completely independently of the patient’s subjective feelings. This method could be used for objective assessing in chemical injuries of the nasal cavity, damage of the peripheral part of olfactory pathway, as well as lesions of central olfactory areas and neurodegenerative diseases associated with anosmia. This method predominantly tests only orthonasal olfaction as well as majority of subjective and objective methods. It is known that retronasal olfaction could bring different results or could lead to activation of different cortex structures. Besides the obvious reasons (polyps, post-traumatic narrowing of the anterior part of the nasal cavity), small changes in cortical processing or other factors as less exposure of the dorsal part of olfactory mucosa could be the reason. Complexity, high requirements for accurate test preparation, as well as time and cost of this examination are the reasons why the use of this method is currently limited to the experimental level. However, given the possibilities this method offers and its wide interdisciplinary utilization, it certainly makes sense to strive for its further development and integration into clinical practice in the future, especially for its importance in the early diagnosis of, e.g., neurodegenerative diseases. In Europe, research using objective olfactometry is mainly carried out by prof. Dr.med. Thomas Hummel, prof. Dr.med. Bertold Renner and Dr. Phillippe Rombaux, Ph.D. After a short internship with the previously mentioned prof. Thomas Hummel, our clinic succeeds in putting the objective olfactometry examination into practise using Burghart OL 024 olfactometer (see Figure 7). Together with our team, we performed olfactory tests on a group of healthy probands whose results will be published. We consider the German study by Stuck et al., who reported that, based on electrophysiological data obtained in a large sample size, the results established an age-related loss of olfactory and trigeminal function, which appears to be almost linear. Further, the results emphasized that responses to chemosensory stimuli are related to sex, while the side of stimulation did not play a major role in the paradigm used [38].

Idiopathic olfactory loss (IOL) accounts for a sizable fraction of olfactory dysfunction, but very little is known about its aetiology and electrophysiological changes in the olfactory pathway. Liu et al. published an interesting study in which they reported that reliability of OERPs is comparable to auditory and visual ERPs. Thus, OERPs might be a more sensitive measure of olfactory dysfunction than psychophysical tests, especially for early diagnosis of neurodegenerative diseases. OERPs and olfactory pathway MRI appear to provide useful information for evaluating patients with idiopathic olfactory loss (IOL) [39].

Miao et al. reported that closed head injury could induce anosmia; the severity extent, injury site, and subsequent consciousness are related to the olfaction. OERP is the gold standard for olfactory subjective examination; MRI could indicate the lesions on the olfactory pathway and reflect the possibility of detectable OERPs [40].

## 8. Personal Experience

Our team has available a Sniffin‘Sticks test from Burghart (Figure 6) [41,42] and the objective olfactometer Burghart OL 024 (Figure 7) for olfactory testing. For examination of nasal patency, we have available endoscopic systems and the Rhinomanometry Otopront Rhino-sys (Figure 5). We have personal experience with olfactory testing in patients with chronic rhinosinusitis (Figure 3) [16], cancer of the paranasal sinus, olfactory meningioma (Figure 4), pituitary adenoma [42], Parkinson’s disease, and after COVID-19. The results of olfactory testing in these patient groups will be published separately in the future. Figure 10 shows our personal strategy for the indication of olfactory tests. (Figure 10).

## 9. Leading Outlook for the Future


-Detailed research on the objective assessment of olfactory loss in patients after COVID-19.-Detailed research on the objective assessment of olfactory loss in patients with Parkinson’s disease.-Leading outlook for the future: research on the objective assessment of olfactory loss in patients with chronic rhinosinusitis (CRS) and nasal polyps (NP)—correlation of the microbiome in nasal polyposis and chronic rhinitis with olfactory disorders using objective olfactometry [16,30,41].-Leading outlook for the future: in collaboration with neurosurgery—detailed research on the objective assessment of olfactory loss in patients before/after pituitary adenoma endoscopic surgery [42].-Leading outlook for the future: in collaboration with radiodiagnostics—detailed description of MRI findings of the olfactory bulb in IOL and traumatic olfactory loss [39,40].


## 10. Conclusions

Impaired olfactory function is a problem that affects an individual’s life at many levels. These are not just personal or social limitations. Loss of smell also represents a safety risk, for example, if an individual cannot smell the smoke of burning objects or escaping natural gas. Mercaptan is added to natural gas as a safety precaution to indicate its leakage or ingestion of liquids in bottles by mistake. It is increasingly evident that olfactory loss is a side effect of some neurodegenerative diseases and appropriate screening could lead to early diagnosis. At present, olfactory loss is also widely discussed in connection with COVID-19 disease. It is one of the first symptoms and it persists for several weeks or months after the disease. Leading outlook for the future is a call for new projects concerning the correlation of the microbiome in nasal polyposis and chronic rhinitis with olfactory impairment using objective olfactometry and research on the objective assessment of olfactory loss in patients before/after pituitary adenoma endoscopic surgery [30,41,42].

Examination of this sense should therefore not be neglected. Objective olfactometry appears to be the method with the greatest potential, and further research and data collection in practice could lead to its routine use across medical disciplines soon.

## Figures and Tables

**Figure 1 jcm-11-06628-f001:**
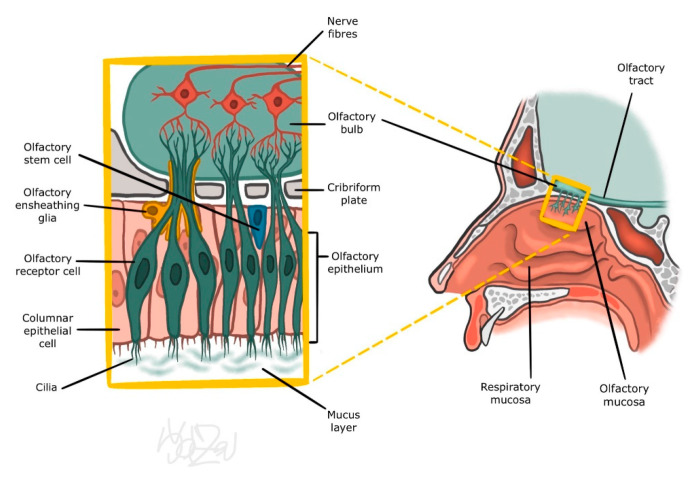
Olfactory epithelium (from co-author’s archive).

**Figure 2 jcm-11-06628-f002:**
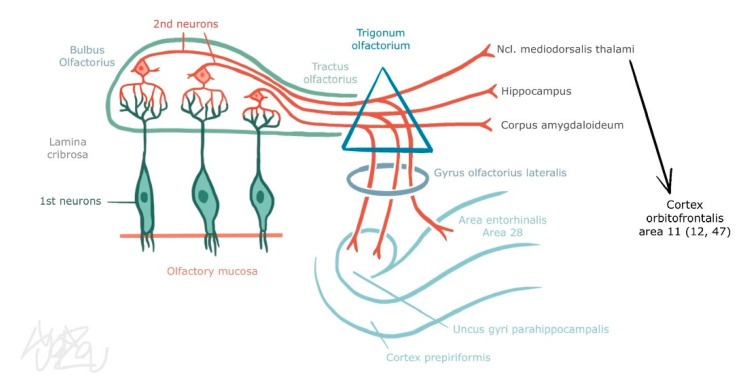
Olfactory pathway (from co-author’s archive).

**Figure 3 jcm-11-06628-f003:**
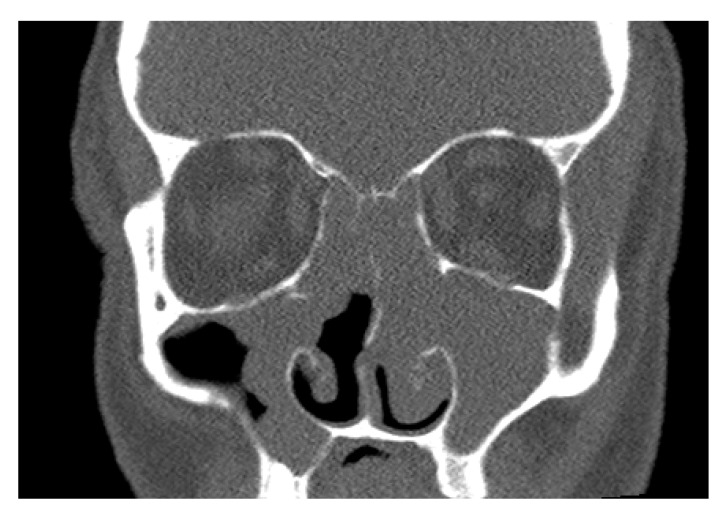
The computerized tomography (CT) coronary scan—chronic rhinosinusitis with nasal polyps. Male, 54 years old (from co-author’s archive).

**Figure 4 jcm-11-06628-f004:**
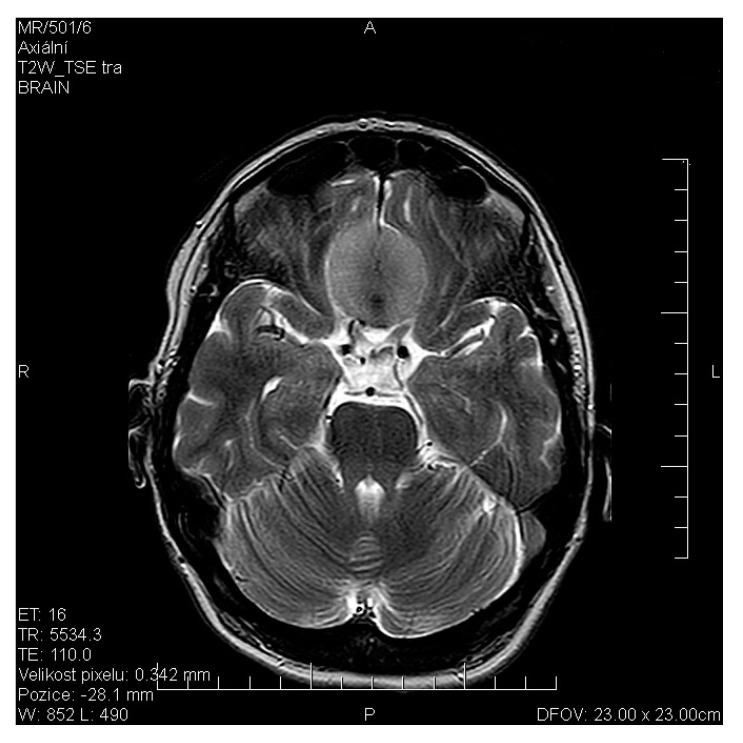
Magnetic resonance imaging (MRI) axial scan—olfactory meningioma—female, 48 years old (from co-author’s archive).

**Figure 5 jcm-11-06628-f005:**
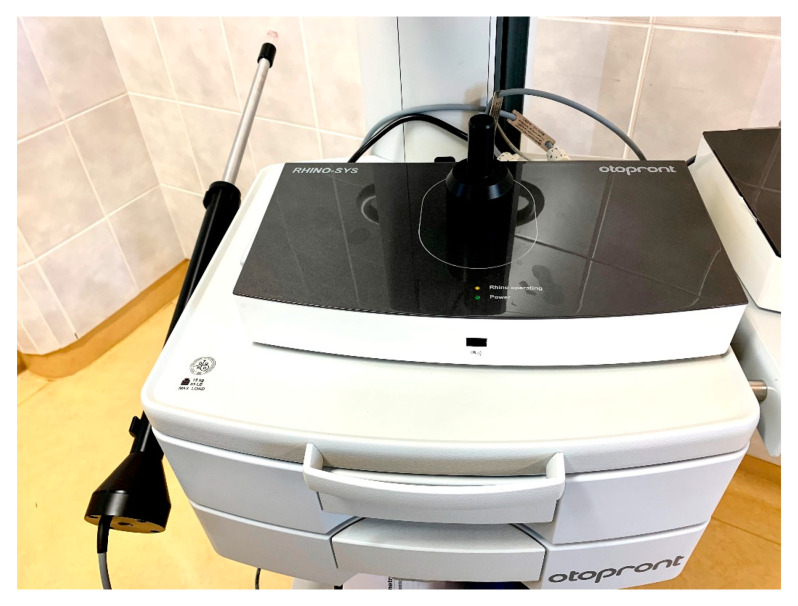
Rhinomanometry Otopront Rhino-sys (from co-author’s archive).

**Figure 6 jcm-11-06628-f006:**
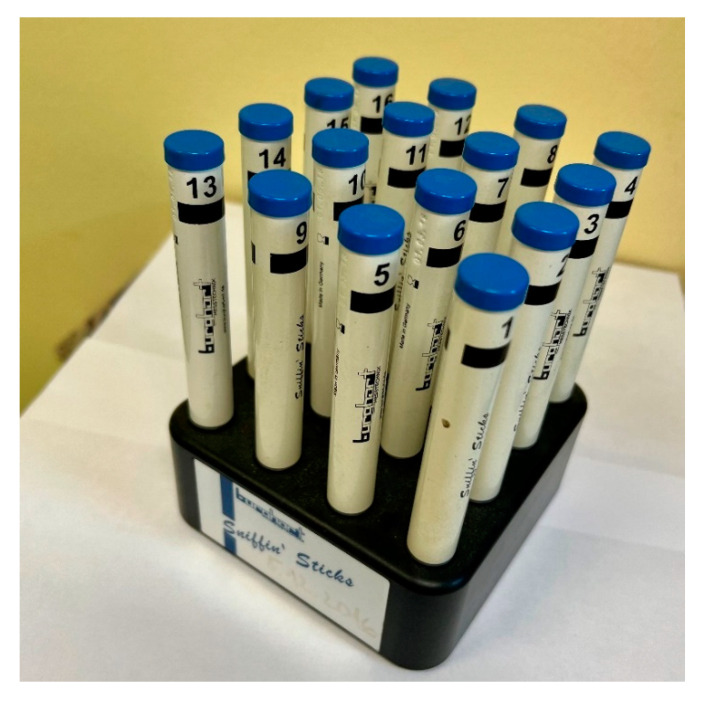
Sniffin ‘Sticks test from Burghart (from co-author’s archive).

**Figure 7 jcm-11-06628-f007:**
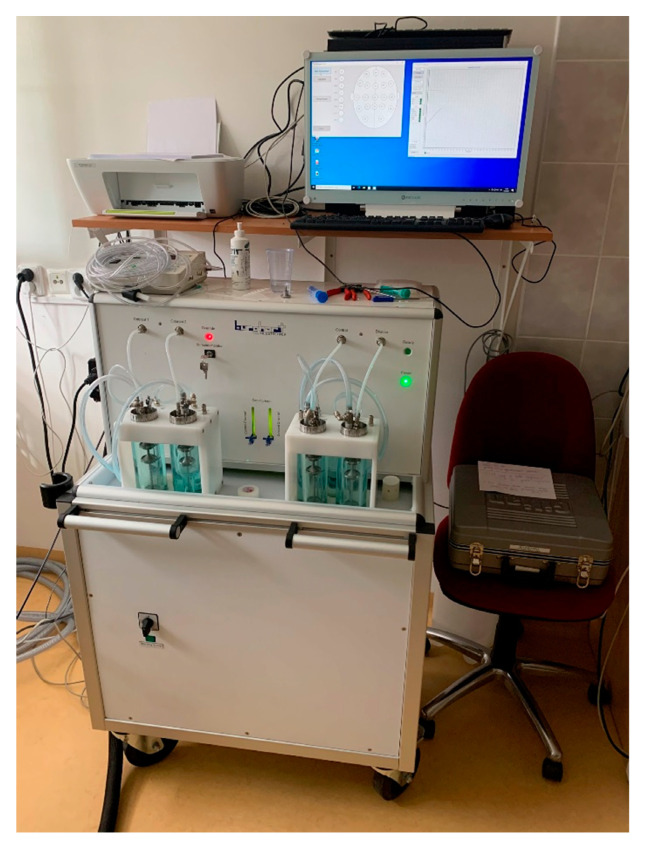
The olfactometer OL 024 Burghart (from author’s archive).

**Figure 8 jcm-11-06628-f008:**
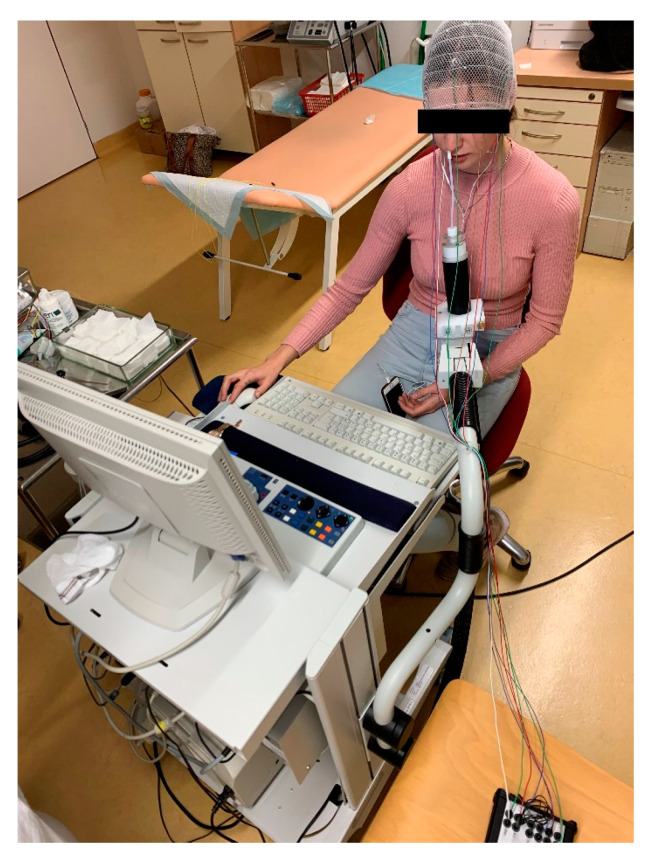
Objective olfactometry (from co-author’s archive).

**Figure 9 jcm-11-06628-f009:**
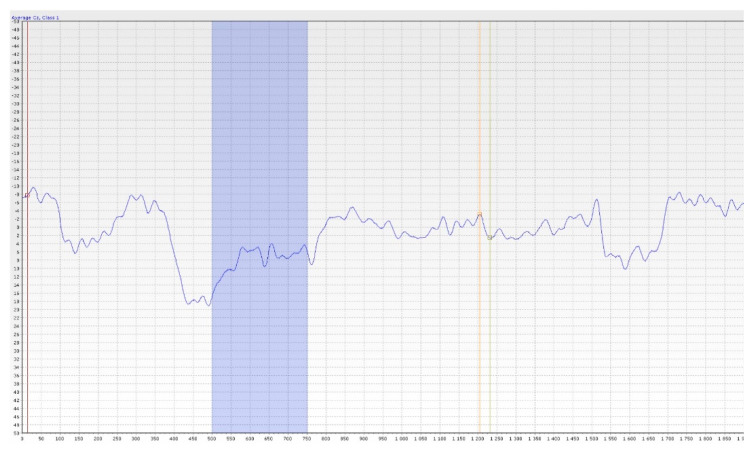
The curve of olfactory event-related potentials (OERP), Female 38 years old, healthy proband (from co-author’s archive).

**Figure 10 jcm-11-06628-f010:**
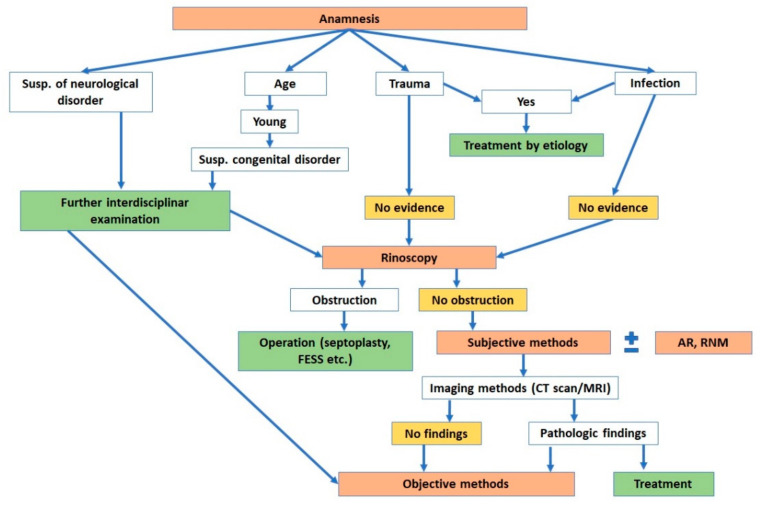
Our strategy for the indication of olfactory tests.

**Table 1 jcm-11-06628-t001:** Disorders of smell by symptomatology.

Quantitative	Qualitative
Hyposmia	Parosmia
Anosmia Hyperosmia	Phantosmia Specific anosmia

**Table 2 jcm-11-06628-t002:** Disorders of smell by etiopathogenesis.

Conductive	Sensorineural
Loss of contact with olfactory region	Disorders of olfactory epithelium
Ventilation failure	Disorders of the olfactory pathway

**Table 3 jcm-11-06628-t003:** Olfactory assessment methods.

Subjective	Objective
Sniffin‘ Sticks Test Odorized Markers Test (OMT)	Objective olfactometry
University of Pennsylvania Smell Identification Test (UPSIT)	

## Data Availability

Not applicable.

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
