# Peer review of "Olfactory Evaluation in Clinical Medical Practice"

_jcm, 2022, doi:10.3390/jcm11226628_

Round 1
Reviewer 1 Report
The review „Olfactory evaluation in clinical medical practice” highlights the role of olfactory testing in clinical practice. The authors also highlighted possible recommendations in using those tests.
I suggest a few major revisions. Comments are made below regarding the article.
- The authors should extend and clarify in the introduction the aim and objectives of the systematic review.
- Is needed a flow chart to clarify the strategy when olfactory test is indicated to be used.
- If personal experience is included, make it as a different chapter.
- Discussion should be added as a different chapter.
- References should be modified according to the journal requests for publication.
Author Response
Dear Reviewer,
Thank you for your recommendation.
We have accepted all of your suggestions and incorporated them into the manuscript upgrade of Revision I. The changes are marked in yellow and red in the manuscript. We have also added new references.
Sincerely
on behalf of the author team
Richard Holy
Corresponding co-author

Author Response
Dear Reviewer,
Thank you for your recommendation.
We have accepted all of your suggestions and incorporated them into the manuscript upgrade of Revision I. The changes are marked in yellow and red in the manuscript. We have also added new references.
Sincerely
on behalf of the author team
Richard Holy
Corresponding co-autho

Reviewer 3 Report
The reviewers received a comprehensive review of the evaluation of OLFACTORY in clinical practice. The overall organization of the chapters, starting with 1. an introduction, 2. anatomy of the nasal mucosa, 3. physiological description, and 4. clinical classification, 5, 6, is clear and understandable. The structure of the book is easy to understand. This review should be easy to understand not only for clinicians specializing in otolaryngology, but also for physicians not involved in otolaryngology. This is because odor dysesthesia is a disorder that has become very major since the coronary infection, but more than that, it is a symptom that is considered to be more frequent in daily practice. With this in mind, the reviewers suggest the following basic content modifications to the authors.
Major1
Visual impairment is classified as a visual impairment, and hearing impairment is classified as a hearing impairment. Is an impairment of the olfactory function or of the sense of smell then classified as a disability? What are the views of WHO and related societies on this point? I propose to add this question to the introduction.
Major2
In Introduction L39-40 the authors state that odor disorders are a common problem in otolaryngology clinics, especially odor loss and problems associated with upper airway inflammation and its disorders. What is the frequency of congenital or acquired odor disorders? Please add any data on frequency statistics.
Major3
2. in the chapter on nasal mucosa there is a description of the text. Is it possible to add illustrations to make it clearer?
Major4.
Chapters 4 and 5 describe the classification of OLFACTORY DISORDERS and the diseases. Can you provide MRI images or photographs as clinical examples of the various disorders?
Below are some minor modifications.
Minor 1
What is the frequency of olfactry disorder in children? What is the impression that it is more common in the elderly?
Minor 2
How are congenital odor disorders such as Kalman's syndrome and olfactry bulb dysplasia diagnosed?
Minor 3
You mention testing in the latter part of the question. At what age is it possible to start testing for odors?
Minor 4
The aftereffects of novel coronavirus-related odors are a worldwide problem. Can you add your approach to these cases and the latest literature?
Best regards,
Dr. Reviewer
Author Response

(The authors gave the same response as above.)

Round 2
Reviewer 1 Report
The review „Olfactory evaluation in clinical medical practice” highlights the role of olfactory testing in clinical practice. The authors also highlighted possible recommendations in using those tests.
I suggest a few minor revisions of spelling and punctuation.
Reviewer 2 Report
The author has made detailed revisions to the article, especially the outlook of future olfactory evaluation methods, which will inspire readers.
Also, the author adds figures corresponding to the article's content to introduce the article's relevant content better.
In addition, the author also provides appropriate references to the important expositions in the article, which makes the paper more scientific and rigorous.
Finally, please check the English text for spelling and typographical errors. I agree to the publication of this article.
Reviewer 3 Report
The reviewers read the authors' revised manuscript with great interest.
Thank you for your clear comments and corrections to all four difficult major and four minor points raised by the reviewers.
In particular, the reviewers appreciate that the authors have added radiological images and anatomical figures, which dramatically increase the value of this review.
Also, the authors have provided us with appropriate literature citations, making the paper more scientifically based.
Finally, please check the English text for spelling and typographical errors.
The reviewers highly commend this paper.
Please make revisions with the input of other reviewers.
The reviewers strongly hope that this review paper will be accepted by JCM.
Best regards,
Dr. Reviewer